# Priority-Based Resource Allocation Optimization for Multi-Service LoRaWAN Harmonization in Compliance with IEEE 2668

**DOI:** 10.3390/s23052660

**Published:** 2023-02-28

**Authors:** Yang Wei, Kim Fung Tsang, Wenyan Wang, Morgana Mo Zhou

**Affiliations:** Department of Electrical Engineering, City University of Hong Kong, Hong Kong 999077, China

**Keywords:** priority-based resource allocation, harmonization, LoRaWAN, IEEE 2668

## Abstract

Given the advantage of LoRaWAN private networks, multiple types of services have been implemented by users in one LoRaWAN system to realize various smart applications. With an increasing number of applications, LoRaWAN suffers from multi-service coexistence challenges due to limited channel resources, uncoordinated network configuration, and scalability issues. The most effective solution is establishing a reasonable resource allocation scheme. However, existing approaches are not applicable for LoRaWAN with multiple services with different criticalities. Therefore, we propose a priority-based resource allocation (PB-RA) scheme to coordinate multi-service networks. In this paper, LoRaWAN application services are classified into three main categories, including safety, control, and monitoring. Considering the different criticalities of these services, the proposed PB-RA scheme assigns spreading factors (SFs) to end devices on the basis of the highest priority parameter, which decreases the average packet loss rate (PLR) and improves throughput. Moreover, a harmonization index, namely HDex, based on IEEE 2668 standard is first defined to comprehensively and quantitively evaluate the coordination ability in terms of key quality of service (QoS) performance (i.e., PLR, latency and throughput). Furthermore, Genetic Algorithm (GA)-based optimization is formulated to obtain the optimal service criticality parameters which maximize the average HDex of the network and contribute to a larger capacity of end devices while maintaining the HDex threshold for each service. Simulations and experimental results show that the proposed PB-RA scheme can achieve the HDex score of 3 for each service type at 150 end devices, which improves the capacity by 50% compared to the conventional adaptive data rate (ADR) scheme.

## 1. Introduction

With the development of the Internet of Things (IoT), over 75 billion IoT devices are expected to be connected, creating about USD 11 trillion in economic benefits by 2025 [1]. Conventional short-range wireless communication technologies (e.g., ZigBee, Bluetooth, and Wi-Fi, etc. [2]) cannot support such large-scale IoT deployments, so long-range wireless communication technologies have emerged to advance global IoT progress. LoRaWAN is one of the most promising long-range wireless communication technologies, which shows its superiority of simplicity and flexibility of private network deployment over others (e.g., narrowband IoT [3], Sigfox [3], etc.). Given the advantages of LoRaWAN, plenty of LoRaWAN-based applications [4,5,6,7,8,9] have been developed to facilitate the realization of Industry 4.0 and smart city blueprints, such as smart lighting [4], indoor localization [6], smart healthcare [8], etc.

However, as LoRaWAN expands and the number of applications increases, LoRaWAN suffers from harmonization issues pertaining to limited channel resources, uncoordinated network configuration, and scalability issues. First, serious LoRaWAN resource competition may occur due to limited unlicensed frequency bands. The unlicensed band of LoRaWAN provides users with the freedom and flexibility to deploy private networks without permission, but the competition for channel resources becomes more and more intense among increasing number of users. Based on Aloha, LoRaWAN end devices transmit packets on a randomly selected channel, which further exacerbates channel contention. Once channel resources are fully occupied by noncritical services, safety-related critical services will not be accessible with disastrous consequences. As such, it is necessary to prioritize the allocation of resources to critical services. Second, there is no standard or scheme to effectively coordinate multi-user network configurations (e.g., spreading factor (SF), etc.). LoRaWAN supports six SF values (i.e., SF7 to SF12) for balancing the data rate and communication range. SF7 enables the fastest data rate and shortest communication range, while SF12 supports the longest communication range with the slowest data rate. Without coordination, end devices transmit packets with specific predefined SF values by users. When excessive end devices select the same SF to transmit packets on the same channel at the same time, packet collision occurs. Third, scalability issues arise. With the increasing number of connected end devices, packet collision would be more serious. The overload of using the same SF values will increase, the packet collision probability will increase, and more critical and noncritical services will be affected. These harmonization issues would directly degrade LoRaWAN QoS performance, including high packet loss rate (PLR), high latency, and low throughput. Moreover, degraded QoS performances reflect low network use efficiency, reducing the number of covered end devices.

To facilitate LoRaWAN harmonization, the most effective way is to make a reasonable allocation of resources to ensure QoS of all services in the network and improve the entire network capacity. A standard adaptive data rate (ADR) scheme supported by legacy LoRaWAN is one of the most popular resource allocation methods. In the standard ADR scheme [10], the network server tries to assign the lowest possible SF value to each end device on the basis of connectivity between the end device and the gateway. To be specific, the network server determines the SF value for the given end device based on the last 20 packets received from the end device. If the maximum signal-to-noise-ratio (SNR) among the last 20 received packets meets the requirement of receiving sensitivity of the lowest possible SF, then this SF is assigned to the end device to improve data rate and reduce transmission time. The standard ADR scheme provides a feasible dynamic SF assignment approach and alleviates collision to a certain degree, but it still needs to be improved. Based on the standard ADR scheme, some modified ADR schemes were studied [10,11,12,13]. Some QoS performances were improved in their simulated or experimental scenarios. However, only LoRaWAN uplinks or single types of traffic are considered in these studies. When multiple types of services are deployed in the network, the QoS performance announced by previous works may not be guaranteed in the coexistence environment. Some resource allocation schemes [14,15,16,17] use Artificial Intelligence (AI) algorithms to solve harmonization issues. Nevertheless, they may cause large latency due to great computation complexity, further suffer from difficulties with practical implementation. Other assistance methods [18,19], such as increasing retransmission times, increasing the number of gateways, and increasing the number of antennas, have improved LoRaWAN QoS performances but at the expense of increased energy consumption and costs. 

To address these challenges, a LoRaWAN harmonization strategy is proposed to coordinate multi-service networks based on priority levels. In this paper, IoT application services are classified into three main categories: safety, control, and monitoring. Considering LoRaWAN’s features, the three kinds of services are modeled as three typical LoRaWAN traffics—Class A non-periodic uplink transmissions with acknowledgement (ACK), automated Class C downlink transmissions with ACK, and Class A periodic unconfirmed uplink transmissions, respectively. Meanwhile, typical use cases with reasonable duty cycles are also illustrated accordingly. Considering the different criticalities of these services, a priority-based resource allocation (PB-RA) scheme is developed to assign SFs on the basis of the highest priority parameter, which decreases the average packet loss rate (PLR) and improves the throughput. Given the different performance requirements of different services, a harmonization IDex based on IEEE 2668 standard [3] is first defined to comprehensively and quantitively evaluate the coordination ability in terms of key quality of service (QoS) performance (i.e., PLR, latency and throughput). Moreover, genetic algorithm (GA)-based optimization is formulated to find optimal service criticality parameters to maximize the average HDex of the network, which contributes to a larger capacity of end devices while maintaining the HDex threshold for each service. 

The key achievements of the paper are as follows:(1)A priority-based resource allocation (PB-RA) scheme is proposed to assign spreading factors (SFs) for different services (i.e., safety, control and monitoring services) on the basis of the highest priority factor, which decreases average packet loss rate (PLR) and improves throughput.(2)A harmonization IDex (HDex) based on IEEE 2668 is first defined to comprehensively and quantitively evaluate the coordination ability in terms of key QoS performance (i.e., PLR, latency and throughput).(3)Genetic Algorithm (GA)-based optimization is formulated to find optimal service criticality parameters to maximize the average HDex of the network, which contributes to a larger capacity of end devices while maintaining the HDex threshold for each service.

This paper is organized as follows: Section 2 discusses related works. Section 3 presents the PB-RA scheme. Section 4 introduces the proposed HDex with IEEE 2668 standard. Section 5 presents GA-based optimization for PB-RA. Section 6 reveals the results and analysis. The conclusion is drawn in Section 7.

## 2. Related Work

In recent years, researchers and developers have been exploring effective resource allocation schemes to achieve LoRaWAN harmonization. The existing approaches in the state of the art are generally grouped into extended ADR methods, AI-based methods, and additional assistance methods.

The standard ADR scheme was designed in legacy LoRaWAN with the goals of increasing network capacity and extending battery life [10]. In ADR, the SF is allocated to a specific end device dynamically through measuring SNR between the end device and the gateway. Based on ADR, a lower SF can be assigned to an end device that is close to a gateway, leaving other SFs for connecting end devices far away from the gateway. Through this SF assignment method, ADR increases the number of covered end devices in the network. Moreover, battery life is also prolonged with assisted transmission power assignment. By far, ADR scheme is still the most widely implemented approach in the LoRaWAN market. However, ADR showed its deficiencies in the performance of critical services and varied physical channel environments. To advance the ADR scheme, the authors of [10] proposed a simple modification, namely ADR+, where the SF is assigned according to average SNR rather than maximum SNR. With this tiny change, ADR+ showed an improved delivery ratio in a noisy environment. The authors in [11] extended the classic ADR scheme to ExpLoRa-SF. This strategy aimed to distribute SF values evenly to end devices in the network based on their radio conditions. The results of ExpLoRa-SF are verified by simulation, showing a better robustness in different operating conditions. The authors of [13] found that different SFs exhibit different capacities of end devices given the same specific collision probability for each SF. Hence, the CA-ADR algorithm was proposed to distribute SF values to an end device based on the targeted collision probability, which outperformed the standard ADR scheme in the SF selection. To support mobile end devices, another paper [12] proposed the LR-ADR and LR + ADR mechanisms. The enhancement of packet delivery ratio and energy consumption for mobile end devices was shown through simulations.

With the goal of improving LoRaWAN QoS performance, many researchers developed resource allocation schemes using strong AI algorithms. The authors of [16] proposed a smart SF assignment algorithm using support vector machines and decision tree classifier machine-learning techniques. The promising performance of the packet delivery ratio was verified by simulation. The authors of [17] developed a SF allocation scheme-based K-means algorithm to equalize the traffic load among SF channels. Through simulation with different traffic loads, it presented a significant enhancement in terms of data extraction rate. The authors of [15] proposed a fair-based SF allocation scheme to maximize the LoRaWAN system throughput using game theory. Their simulation results showed an improvement in the packet delivery ratio. Moreover, a many-to-one matching algorithm was presented by the authors of [14] to allocate SF to each end device. This algorithm was evaluated to achieve an enhancement in throughput and packet delivery rate.

Additional assistance methods for LoRaWAN resource allocation were also studied. The authors of [19] explored a new resource allocation scheme with the assistance of retransmission, namely R-ARM. Through operating both on end device and network server sides, the R-ARM system greatly improved the packet success ratio and convergence period, and lowered the energy consumption. Employing message replication and gateways with multiple receive antennas were also explored by the authors of [18] to improve LoRaWAN performance with time and antenna diversity.

The above resource allocation approaches illustrated the effectiveness of different QoS performances (e.g., PLR, latency, and throughput, etc.). Nevertheless, these existing works only consider LoRaWAN uplinks or a single type of service. They may not be applicable to a multi-service LoRaWAN system. In fact, multiple types of services are usually implemented in one LoRaWAN system to realize various applications. Hence, coexistence performance is essential to study. Moreover, different priority levels of different services also need to be considered so that both critical services and noncritical services can meet their own requirements. 

## 3. Priority-Based Resource Allocation (PB-RA) Scheme

### 3.1. Priority-Based Service Model in LoRaWAN

In general, IoT application services can be classified into three main categories: safety, control, and monitoring [20]. IoT services in each category have different functionalities and criticalities. The safety category refers to emergency messages when the alarms are triggered, which usually require an immediate response. The control category is command messages sent from operators to end devices within a short time. The monitoring category usually represents the periodic sensor measurements from end devices to the network server. In most realistic scenarios, the three categories’ services coexist and collaborate. It is obvious that the criticality order of the three categories is $pr_{\mathrm{safety}}>pr_{\mathrm{control}}>pr_{\mathrm{monitoring}}$ from the highest to the lowest. In this paper, three categories of IoT services in LoRaWAN are considered. Taking LoRaWAN’s features, these services are illustrated as follows:(1)Safety Service: Safety services depend on the occurrence of sporadic alarm events and timely emergency response. It can be modeled as Class A non-periodic uplink transmissions with acknowledgement (ACK) in LoRaWAN.(2)Control Service: Control services rely on remote commands from the network server with quick response, which is modeled as automated Class C downlink transmissions with ACK in LoRaWAN.(3)Monitoring Service: Monitoring services are periodic sensor measurements, which can be modeled as Class A periodic unconfirmed uplink transmissions in LoRaWAN.

### 3.2. Resource Allocation Scheme

LoRaWAN is one of the most popular LPWA technologies, which provides a kilometers-level transmission range and only uA-level power consumption. Given these advantages, the number of LoRaWAN end devices increases exponentially, which also induces more interferences, leading to serious packet collision. In addition, the limited channel resources due to unlicensed bands further exacerbate resource contention, which may result in critical messages (e.g., healthcare alarms) not being transmitted and served in time. To address these challenges, a priority-based resource allocation scheme is proposed to prioritize emergent packets and mitigate overall packet collision.

In LoRaWAN, there are six SFs (i.e., SF = 7,8,9,10,11,12) per channel to choose from to balance the data rate and transmission range. Signals transmitted with different SFs show orthogonality [1]. Thus, in this paper, PB-RA exploits this orthogonality feature to allocate SF values to each end device reasonably to reduce the collision probability of packets. Based on the previous work [13], the theoretical packet success probability $\hat{p}\left(\mathrm{SF}\right)$ in the MAC layer for the end devices using the specific SF value can be formulated as
(1)$$\hat{p}\left(\mathrm{SF}\right)={\left(1-\frac{\mathrm{ToA}\left(\mathrm{SF}\right)}{T}\right)}^{2\left(n_{max}\left(\mathrm{SF}\right)-1\right)},\;\mathrm{SF}=7,8,9,10,11,12\;$$
where ${\mathrm{ToA}}_{\mathrm{SF}}$ is the time on air of the packet using SF. $n_{\mathrm{SF}}$ is the number of end devices using the SF. T is the time duration (here, T is set as 1 min). Based on Equation (1), the maximum number of end devices $max\;n_{\mathrm{SF}}$ that can use the SF can be derived.

Unlike previous algorithms which assign SF values to end devices evenly, the proposed PB-RA considers different priority levels and assigns SFs with the order of priority levels to harmonize the entire LoRaWAN network.

In PB-RA, the priority service function is designed considering the service criticality and the contribution of service offered throughput to the network, which not only ensures service reliability but also improves the network use rate. Thus, the formulation of the priority factor $F_{nk}$ for $n^{th}$ end device with $k^{th}$ service is defined as:(2)$$F_{nk}=\omega_{n}\cdot pr_{k}=\frac{O_{n}}{\sum_{n}^{N}O}\cdot pr_{k},\;n=1,2,\ldots N,\;k=1,2,\ldots K\;$$
where N is the total number of end devices in the network and K is the total number of services in the network (i.e., K=3). $pr_{k}$ is the service criticality factor of $k^{th}$ service, and $\omega_{n}$ is the contribution factor of $n^{th}$ end device’s offered throughput to the network. To be specific, $\omega_{n}$ can be calculated as the ratio of $n^{th}$ end device offered throughput $O_{n}$ to the total offered service throughput of the entire system $\sum_{n}O_{n}$. We assume that $n^{th}$ end device average generates a packet of payload $B_{n}$ [bytes] within a time period T [s]. In this paper, T=1 min. Hence, the $n^{th}$ end device offered throughput $O_{n}$ can be calculated as the number of bits per second generated in a network as follows:(3)$$O_{n}=\frac{8\cdot B_{n}}{T}\;$$

Based on the defined priority factor $F_{k}$, the PB-RA scheme works as shown in Figure 1. In the initial, $\hat{p}\left(\mathrm{SF}\right)$ for all SFs start from the best value as 1. $F_{nk}$ is calculated according to Equation (2) for each end device and it is embedded in the LoRaWAN frame during the end device configuration phase. Then, the minimum possible SF value for each end device *n* can be derived when the ${\mathrm{SF}}_{min}\left(n\right)$ satisfies $\mathrm{RSSI}\left(n\right)\geq {\mathrm{RSSI}}_{min}\left(\mathrm{SF}\right)$. RSSI(n) is the average receiving signal strength by the gateway during T. The minimum sensitivity of each SF ${\mathrm{RSSI}}_{min}\left(\mathrm{SF}\right)$ is listed in Table 1. With the principle of allocating minimum possible SFs for end devices as much as possible, we start allocating from SF 7. The maximum number of end devices that can be assigned with the specific SF value (i.e., $n_{max}\left(\mathrm{SF}\right)$) is calculated according to Equation (1). Next, prioritization for those end devices with same ${\mathrm{SF}}_{min}$ is performed in order of priority. Specifically, sort the end devices with the same minimum SF value from the highest $F_{nk}$ to the lowest $F_{nk}$. In addition, assign this SF value to end devices in the arranged order, as long as the current number of end devices that assigned with this SF (i.e., count(SF)) does not exceed the precalculated threshold $n_{max}\left(\mathrm{SF}\right)$. If the current number of end devices that assigned with this SF reaches the threshold (i.e., $r=count\left(\mathrm{SF}\right)-n_{max}\left(\mathrm{SF}\right)>0$), then for the rest *r* end devices that have not been assigned this SF value, ${\mathrm{SF}}_{min}\left(n\right)={\mathrm{SF}}_{min}\left(n\right)+1$. Otherwise, loop into the next higher SF value. This process is looped until reaching the highest SF 12. If end devices in the network cannot be fully assigned with the preset value $\hat{p}\left(\mathrm{SF}\right)$, then $\hat{p}\left(\mathrm{SF}\right)$ is decremented by 0.01 and the algorithm runs from the beginning as long as $\hat{p}\left(\mathrm{SF}\right)>0.01$. In particular, the proposed PB-RA scheme can work through revising the standard ADR scheme [13] in LoRaWAN. It should be noted that prioritization is the most significant part in ensuring the reliability of critical services. To verify the effectiveness of the designed prioritization process (the red box in Figure 1), the performance of the proposed PB-RA will be compared with the resource allocation (RA) scheme without priority (without prioritization part).

## 4. Harmonization IDex (HDex) with IEEE 2668

LoRaWAN performance evaluation is a tedious work, especially when there are multiple co-existing application services with different QoS requirements. Conventional QoS metrics are unable to reflect the users’ satisfactory degree. Moreover, the average QoS of the network cannot represent the actual harmonization performance with multiple application services. For instance, the network has a high average network QoS, but the safety services show very low QoS, which is usually not accepted by users. To address this challenge, the HDex with IEEE 2668 standard is proposed as a user-centric comprehensive and quantitative indicator for LoRaWAN evaluation. HDex established a relationship between user requirements for each type of service and their QoS performance (i.e., latency, PDR, and throughput). A final HDex score is provided to present the harmonization degree of all LoRaWAN services in simple five levels. It should be noted that HDex mainly focuses on the coordination performance of multiple services in the radio layer. Hence, this paper does not consider the performance metrics (e.g., energy consumption, cost, etc.) that are only related to itself and would not affect the transmission of other end devices.

### 4.1. QoS Metrics

In HDex evaluation, the main QoS metrics, including latency, PLR, and throughput are considered. The definitions of these three QoS metrics for different types of services are presented as follows.


(1)*Latency (D):* Latency is the interval of time between packet generation at the end device (or network server) and packet successful reception at the network server (or end device) with/without ACK. The latency for each type of service is defined as:


(4)$$\begin{array}{c} D_{safety}=D_{UL}+D_{ACK}\\ =(T_{ED}+ToA_{UL}+T_{GW}+T_{GW-NS}+T_{NS})+(T_{NS}+T_{GW-NS}+T_{GW}+ToA_{ACK}+T_{ED}) \end{array}$$(5)$$\begin{array}{c} D_{control}=D_{DL}+D_{ACK}\\ =(T_{NS}+T_{GW-NS}+T_{GW}+ToA_{DL}+T_{ED})+(T_{ED}+ToA_{ACK}+T_{GW}+T_{GW-NS}+T_{NS}) \end{array}$$(6)$$D_{mornitoring}=D_{UL}=T_{ED}+ToA_{UL}+T_{GW}+T_{GW-NS}+T_{NS}$$
where $D_{UL}$ is the time for one complete uplink transmission, which is composed of the time for generating a packet by an end device ($T_{ED}$), the time on air for transmitting an uplink packet on the sub-channel ($ToA_{UL}$), the time for processing the packet by the gateway ($T_{GW}$), the propagation time from gateway to network server ($\;T_{GW-NS}$) and the processing time at the network server ($T_{NS}$). $D_{ACK}$ is the time interval between ACK generation at network server (or end device) and its successful reception at end device (or network server). $ToA_{ACK}$ denotes the time on air (ToA) [1] for transmitting ACK packet. 


(2)*Packet Loss Rate (PLR):* PLR is the ratio of lost packets at the network server for uplinks or at end devices for downlinks. The PLR for each type of service can be expressed as


(7)$$PLR_{safety}=1-\frac{SN_{UL+ACK}}{N_{UL}}\;$$(8)$$PLR_{control}=1-\frac{SN_{DL+ACK}}{N_{DL}}$$(9)$$PLR_{mornitoring}=1-\frac{SN_{UL}}{N_{UL}}$$
where $N_{UL}$ denotes the total number of generated uplink packets at end devices, $SN_{UL}$ denotes the number of successful received uplinks at network server. $SN_{UL+ACK}$ denotes the number of both successful uplink and its corresponding ACK, and $SN_{DL+ACK}$ denotes the number of both successful downlink and its corresponding ACK.


(3)*Throughput (S):* Throughput is the number of bits that that transmitted successfully per second. The throughput for $k^{th}$ service (i.e., safety, control, monitoring) can be presented as


(10)$$S_{k}=\frac{8\cdot B_{k}\cdot N_{k}}{T}\cdot (1-PLR_{k}),\overset{K}{\underset{k=1}{\sum }}N_{k}=N$$
where $B_{k}$, $PLR_{k}$ denote the average payload size and packet loss rate of $k^{th}$ service, respectively. $N_{k}$ is the number of end devices with $k^{th}$ service, the total number of end devices is denoted as N.

### 4.2. HDex Evaluation

Based on the requirements of QoS and measured QoS metrics, HDex uses the difference gap between measured QoS and expected QoS to evaluate the harmonization degree of the LoRaWAN system. The *i*-th QoS metric difference $u_{i}$ between measured and expected value is defined as
(11)$$u_{i}=\frac{\bar{QoS_{i}^{measure}}-QoS_{i}^{expect}}{QoS_{i}^{expect}}\times 100\%$$
where $\bar{QoS_{i}^{measure}}$ is the *i*-th QoS measured value, $QoS_{i}^{expect}$ is the *i*-th QoS expected value. It should be noted that $u_{i}$ is unified to be larger is better, thus some QoS metrics (i.e., latency, PLR) are transferred to be inverse QoS (i.e., $\frac{1}{\bar{D}},\;\frac{1}{\bar{PLR}}$) instead. According to IEEE 2668 standard [3], HDex is classified as five levels consisting of Bad = 1, Poor = 2, Fair = 3, Good = 4, and Excellent = 5, which is shown in Table 2. 

The HDex estimation function for the $k^{th}$ service is as follows:(12)$$HDex_{k}=\frac{\sum_{i=1}^{l}\alpha_{i}\cdot u_{i}}{\Omega }$$
where $u_{i}$ denotes the *i*-th QoS metric differences (i.e., $\Delta \frac{1}{\bar{D}},\;\Delta \frac{1}{\bar{PLR}},\;\Delta \bar{S}$). l is the total number of QoS metric difference between measured value and the expected value (i.e., l=3). $\alpha_{i}$ is the weighting of the *i*-th QoS metric difference. Ω is normalization factor to transfer the $HDex_{n}$ to the specific range [1, 5]. 

The average HDex of the network can be derived as:(13)$$\bar{HDex}=\frac{1}{K}\sum_{k=1}^{K}HDex_{k}\;$$

Based on HDex score, harmonization degree can be expressed clearly. The network with HDex=3 shows that its performances just meet the QoS requirements of users. The higher the HDex score, the better the LoRaWAN network performance, the better the LoRaWAN system harmonization ability.

## 5. Optimization of PB-RA Scheme

To find the optimal service criticality factors $pr_{k}$ in PB-RA scheme, optimization is performed with the objectives of maximizing the average HDex of the network along with maintaining the HDex threshold of each service (i.e., safety, control, monitoring). The optimization problem is formulated as:(14)$$\begin{array}{c} max\;\bar{HIDex}\left(pr_{k}\right)\\ s.t.\;HDex_{k}>HDex_{k}^{threshold} \end{array}$$
where $pr_{k}$ is the service criticality factor of $k^{th}$ service, $\bar{HDex}$ is the average HDex of the network, $HDex_{k}$ is the HDex score of $k^{th}$ service, and $HDex_{k}^{threshold}$ is the HDex threshold of the $k^{th}$ service.

We assume that end devices with the same type of service have the same criticality level. The service criticality factor $pr_{k}$ is defined as a real number with one decimal place within (0,1], thus the priority factor $F_{nk}$ is also normalized into [0, 1] for each type of service (i.e., safety, control, monitoring). Instead of the exhaustive search approach, a genetic algorithm (GA) [1] is used to reduce the iteration times. Figure 2 shows the optimization process of PB-RA scheme. A set of service criticality factors for each type of service $\left\{pr_{1},pr_{2},\ldots pr_{k}\right\}$ is initialized at first, and the priority factor for each end device $F_{nk}$ can be derived using Equation (2). Based on the derived $F_{nk}$, a PB-RA scheme is performed for SF allocation. The QoS metrics of each type of service (i.e., $D_{k},PLR_{k},S_{k}$) can be calculated through Equations (4)–(10). The final network harmonization score $\bar{HDex}$ is the average value of each service’s $HDex_{k}$. The $\bar{HDex}$ is used as a fitness value for GA algorithm evaluation. The population selection, reproduction, and mutation will be iterated until reaching the following stopping criteria: (1) the derived $\bar{HDex}$ is better or equal to the desired value; (2) the number of iterations reaches 1000 times. Upon the optimization process, the optimal service criticality factors for each type of service $\left\{pr_{1},pr_{2},\ldots pr_{k}\right\}$ can be derived.

## 6. Results and Analysis

Simulations are developed with MATLAB to evaluate the QoS performance of the proposed PB-RA scheme. In addition, optimal criticality parameters are derived using the proposed GA-based optimization process. Moreover, experiments are performed to verify the effectiveness. Detailed information on simulation and experimental setup, the performance evaluation of the PB-RA scheme, and the performance test with the derived optimal criticality parameters are presented in the following.

### 6.1. Simulation and Exprimental Setup

In this paper, practical experiments and simulations based on MATLAB are exploited to verify the effectiveness of the proposed PB-RA scheme. An experiment testbed is set up on the university campus, where 150 end devices are uniformly distributed. Three gateways are deployed in the geometric center of the testing area to maximize network coverage. The three types of services (i.e., safety, control, and monitoring) are included in LoRaWAN. For each type of service, two kinds of traffic with different transmission rates are simulated using the same number of end devices. All these traffic types are typical LoRaWAN applications [21,22] and are in compliance with 1% duty cycle limitation. The number of end devices for each service type varies from 10 to 50. The default eight channels are enabled in most experiment scenarios in addition to testing the effect of the number of channels on the LoRaWAN performance. Detailed simulation and experimental parameters are listed in Table 3. 

### 6.2. Performance Evaluation of PB-RA Scheme

The performances (i.e., PLR, latency, throughput, and HDex) of the proposed PB-RA scheme are analyzed and compared to the performances of ADR scheme and RA scheme without priority. 

Figure 3 shows the PLR performance of different resource allocation schemes as the function of the number of end devices in the network. It is obvious that PLR increases with the growing number of end devices in all three schemes due to the increase in collision probability. The ADR scheme presents the worst PLR performance. This is because the ADR scheme always assigns the lowest possible SF (the highest data rate) to all end devices, where most end devices compete for the resources of lower SFs while leaving higher SFs resources. This causes severe collisions among the end devices that are assigned with the same SFs. By contrast, this co-SF collision is taken into account by the RA scheme without priority and PB-RA scheme. In RA without priority and PB-RA schemes, end devices are assigned with different SFs by tracking the former SF allocations for other end devices, which reduces the collision probability and decreases PLR. The PLR of PB-RA scheme is slightly better than that of RA scheme without priority, which is less than 10% when there are 120 end devices in the network.

The throughput performance of different resource allocation schemes as the function of the number of end devices in the network is shown in Figure 4. As the number of end devices increases, the offered traffic increases and the throughput of the network also improves. A decrease is witnessed in the throughput of the ADR scheme when the number of end devices grows from 120 to 150, which may be a result of serious PLR (i.e., 44%) at 150 end devices. Given the relatively low PLR, the throughput of PB-RA scheme and RA scheme without priority performs similarly and is much better than that of the conventional ADR scheme. Figure 5 illustrates the throughput performance of 150 end devices with different numbers of available channels, which indicates the impact of the number of channels on resource allocation schemes. We can see that the total throughput of the network is almost proportional to the number of available channels in all three schemes, which shows a similar trend to the pure LoRaWAN. Thus, the three schemes can be applied to all scenarios with different numbers of channels. 

Figure 6 illustrates the latency performance of different resource allocation schemes. According to Equations (4)–(6), latency mainly depends on the working status of each end device rather than the number of end devices in the network. The simplest ADR scheme has the lowest average latency of 1.28 s. Since the addition of statistical work on the number of devices assigned to each SF, the latency of the RA scheme without priority is 1.58 s, which is slightly higher than that of the ADR scheme. Based on the RA scheme without priority, priority sorting is added in the PB-RA scheme with an extra about 100 ms latency, which is still acceptable for the entire network.

Apart from the analysis of conventional performance metrics, HDex based on IEEE 2668 is also evaluated in terms of three resource allocation schemes to present the harmonization performance of multiple services in LoRaWAN. As shown in Figure 7, the average HDex of the network decreases with the increasing number of end devices. The PB-RA scheme has the best harmonization performance. The PB-RA could support 150 end devices with no fewer than 3 of the average HDex score; however, only about 100 end devices are supported in ADR and RA without the priority scheme with the same HDex threshold. This is because SF resources are preferentially allocated to services with higher priority and stricter QoS requirements while meeting the demands of lower-priority services. This priority-based strategy contributes a great deal to the overall harmonization performance (i.e., HDex score). In particular, based on randomly selected priority parameters (i.e., $pr_{\mathrm{safety}}=0.8,\;pr_{\mathrm{control}}=0.5,\;pr_{\mathrm{monitoring}}=0.2$), PB-RA achieves an average HDex score of 3.0 with HDex scores of 2.8, 2.9 and 3.1 for safety, control and monitoring services, respectively, when there are 150 end devices in the network. However, with an average HDex score of 3.0 as the goal, the ADR scheme can support up to 100 end devices, and the RA scheme without priority can support a maximum of 110 end devices. In other words, the PB-RA scheme improves by 50% capacity to the conventional ADR scheme and by 36% capacity than RA scheme without priority. A maximum of 100 end devices in the ADR scheme and a maximum of 110 end devices in the RA scheme without priority can be supported. In other words, the PB-RA scheme increases the capacity by 50% compared with the conventional ADR scheme and increases by 36% compared with the RA scheme without priority. 

### 6.3. Finding Optimal Priority Parameters for PB-RA Scheme

For the proposed PB-RA scheme, priority parameters for different types of services are critical to the final HDex scores. In general, apart from the average HDex score, it is necessary to maintain the HDex of each type of service at or above the HDex threshold. Here, the threshold of HDex is set to be 3, which means that each type of service could achieve its basic QoS requirements (i.e., PLR, latency, and throughput). By exploiting the GA algorithm, the optimal priority parameters of different services at 150 end devices can be derived, as shown in Figure 8. It can be seen that the optimal priority parameters are $pr_{\mathrm{safety}}=0.7,\;pr_{\mathrm{control}}=0.4,\;pr_{\mathrm{monitoring}}=0.3$. The average HDex of all services can reach 3.1 maintaining HDex threshold of each service to be no less than 3. Specifically, $HDex_{\mathrm{safety}}=3.0,\;HDex_{\mathrm{control}}=3.1,\;HDex_{\mathrm{monitoring}}=3.2$. 

## 7. Conclusions

Given the advantages of long range, low power consumption, and private network deployment, a lot of LoRaWAN-based applications have been developed. However, with the increasing number of LoRaWAN applications, packet collision becomes more severe due to limited LoRaWAN resources. To address this challenge, a LoRaWAN harmonization strategy was proposed to coordinate multi-service networks based on priority levels. In this paper, LoRaWAN application services are classified into three main categories, including safety, control, and monitoring. Considering the different criticalities of these services, the PB-RA scheme was developed to assign SFs on the basis of the highest priority parameter, which improves the average PLR and throughput. Moreover, a harmonization IDex based on IEEE 2668 standard was defined to comprehensively and quantitively evaluate the coordination ability in terms of key QoS performances (i.e., PLR, latency and throughput). Furthermore, GA-based optimization is formulated to find optimal service criticality parameters to maximize the average HDex of the network, which contributes to a larger capacity of end devices while maintaining the HDex threshold for each service. Simulations and experimental results showed that the proposed PB-RA scheme can achieve the HDex score of 3 for each service type at 150 end devices, which improves capacity by 50% compared to the conventional ADR scheme and resource allocation scheme without priority.

## Figures and Tables

**Figure 1 sensors-23-02660-f001:**
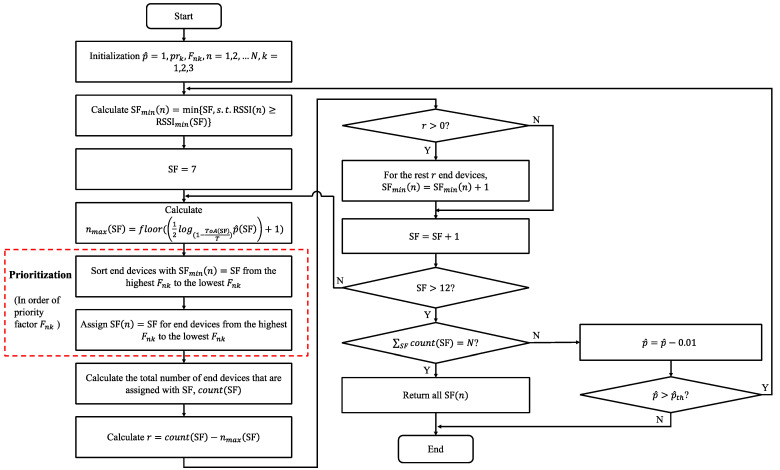
Flow chart of PB-RA scheme.

**Figure 2 sensors-23-02660-f002:**
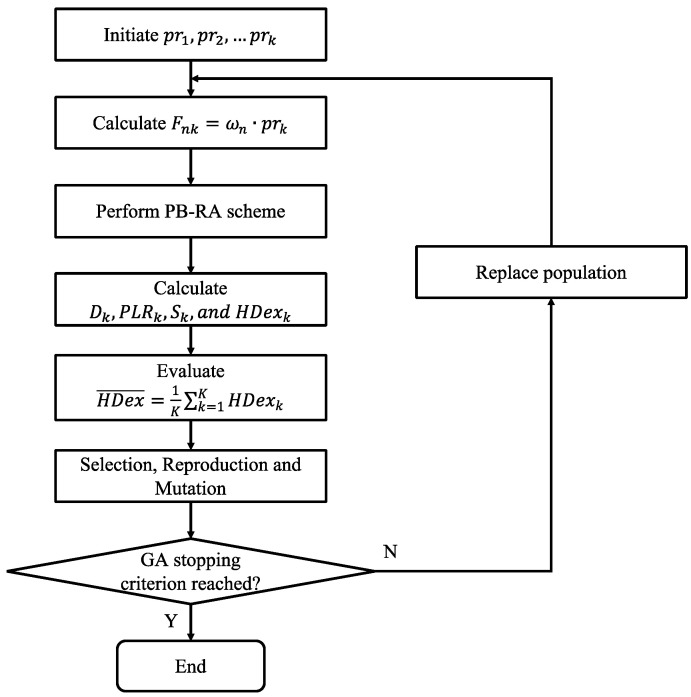
Optimization Process of PB-RA Scheme.

**Figure 3 sensors-23-02660-f003:**
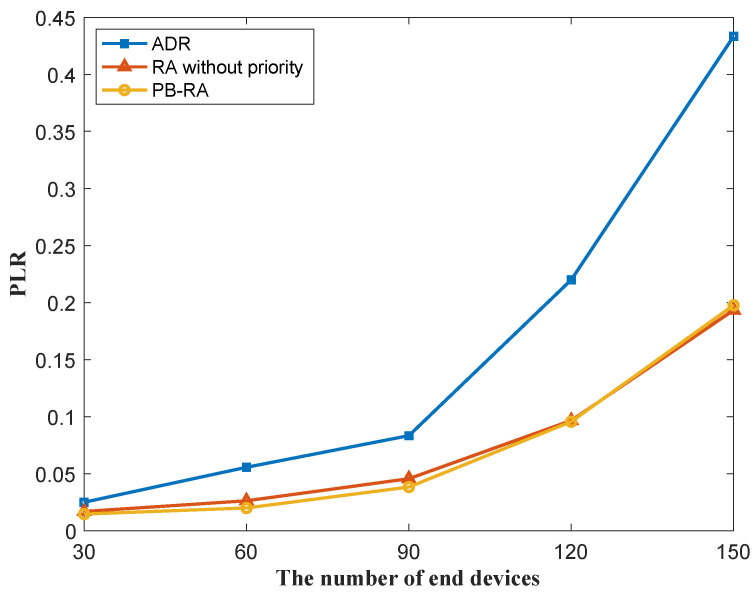
PLR of different resource allocation schemes.

**Figure 4 sensors-23-02660-f004:**
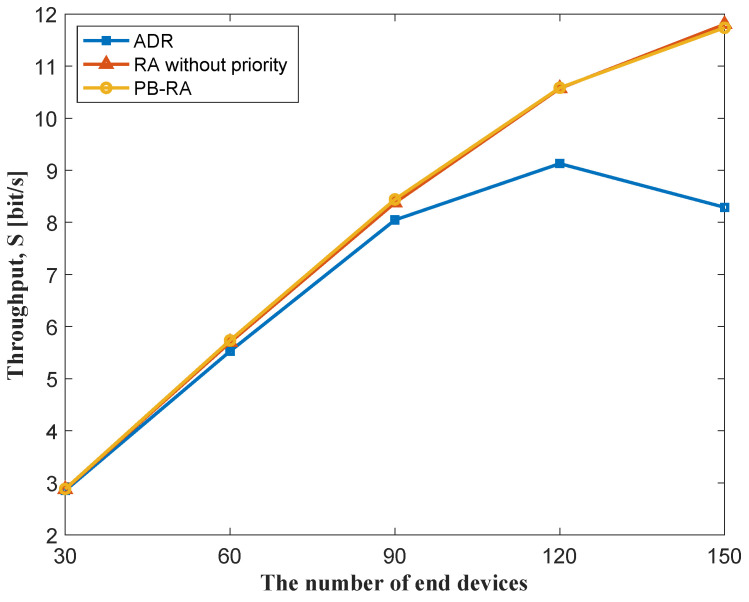
Throughput of different resource allocation schemes.

**Figure 5 sensors-23-02660-f005:**
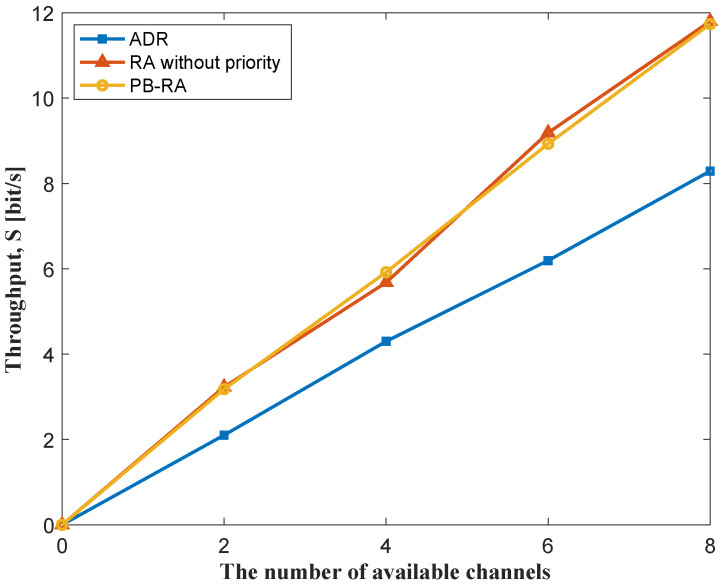
Throughput performance with different number of available channels.

**Figure 6 sensors-23-02660-f006:**
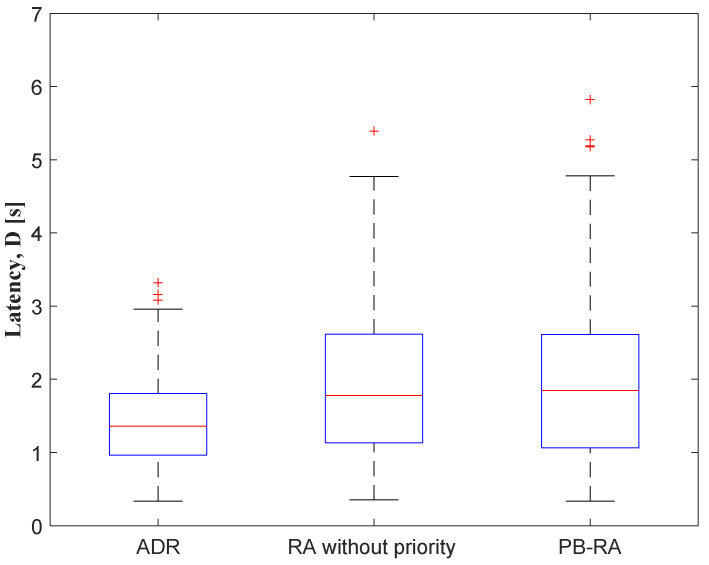
Latency of different resource allocation schemes.

**Figure 7 sensors-23-02660-f007:**
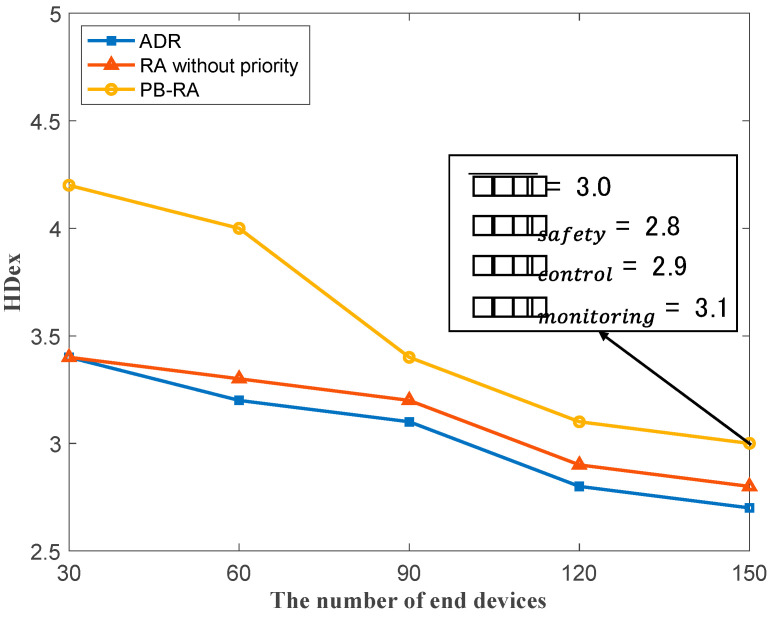
HDex of different resource allocation schemes.

**Figure 8 sensors-23-02660-f008:**
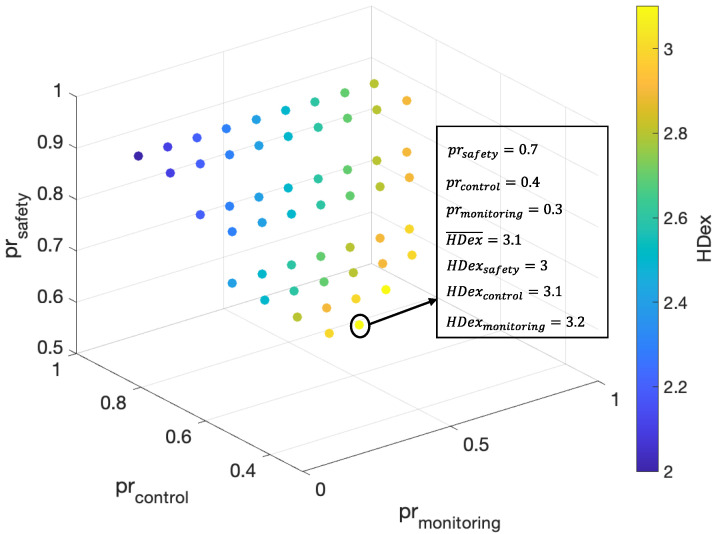
Optimal criticality parameters for PB-RA scheme.

**Table 1 sensors-23-02660-t001:** Minimum possible SF with respect to RSSI sensitivity in LoRaWAN [13].

$RSSI_{min}$ **[dBm]**	−124.5	−127	−129.5	−132	−134.5	−137
$SF_{min}$	7	8	9	10	11	12

**Table 2 sensors-23-02660-t002:** HDex definition for LoRaWAN evaluation.

HDex Level	Description	$\Delta \frac{1}{\bar{D}}$	$\Delta \frac{1}{\bar{PLR}}$	$\Delta \bar{S}$
1	Bad	−200%	−200%	−200%
2	Poor	−100%	−100%	−100%
3	Fair	0	0	0
4	Good	+100%	+100%	+100%
5	Excellent	+200%	+200%	+200%

**Table 3 sensors-23-02660-t003:** Simulation and experimental parameters.

Parameters	Quantity	Configuration
LoRa gateway	3	Multitech Conduit modem with eight channels in AS923
Network server	1	Chirpstack platform in virtual machine
End devices	150	Heltec WiFi LoRa 32 module $N_{1}=N_{2}=N_{3}=10:10:50$
Safety Service model	-	Traffic 1. Alarms [22]: 5 packets/day, payload size = 20 bytes Traffic 2. Smoke detectors [22]: 2 packets/day, payload size = 20 bytes Expected PLR, latency, throughput: 1%, 1 s, 0.006$N_{1}$ bits/s
Control Service model	-	Traffic 1. Smart lighting [21]: 5 packets/day, payload size = 20 bytes Traffic 2. Machinery control [22]: 100 packets/day, payload size = 20 bytes Expected PLR, latency, throughput: 1%, 3 s, 0.096$N_{2}$ bits/s
Monitoring Service model	-	Traffic 1. Smart parking [22]: 60 packets/day, payload size = 20 bytes Traffic 2. Smart metering [21]: 144 packets/day, payload size = 20 bytes Expected PLR, latency, throughput: 10%, 10 s, 0.170$N_{3}$ bits/s

## Data Availability

Not applicable.

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
