# Peer review of "Priority-Based Resource Allocation Optimization for Multi-Service LoRaWAN Harmonization in Compliance with IEEE 2668"

_sensors, 2023, doi:10.3390/s23052660_

Round 1

Reviewer 1 Report

Comments to the Author

The authors have proposed a priority-based resource allocation scheme to assign spreading factors based on the highest priority parameter, decreasing the average packet loss rate and improving throughput. The performance of the proposed method was compared to the typical ADR of LoRaWAN, which improves the performance by 50%. The paper is well written. However, I have the following concerns regarding the improvement of the paper.

1.      The authors need to elaborate on IDex and Hdex in the abstract for the readers.

2.      The article in its present form doesn't meet the recommended length of the paper, which should be more than 16 pages. Therefore, I recommend including another section, "Related Studies," by adding the current research work related to the proposed work. I have mentioned a few here.

[1] Moysiadis, Vasileios, et al. "Extending ADR mechanism for LoRa enabled mobile end-devices." Simulation Modelling Practice and Theory 113 (2021): 102388.

[2] Farhad, Arshad, Dae-Ho Kim, and Jae-Young Pyun. "R-ARM: Retransmission-assisted resource management in lorawan for the internet of things." IEEE Internet of Things Journal 9.10 (2021): 7347-7361.

[3] Slabicki, Mariusz, Gopika Premsankar, and Mario Di Francesco. "Adaptive configuration of LoRa networks for dense IoT deployments." NOMS 2018-2018 IEEE/IFIP Network Operations and Management Symposium. IEEE, 2018.

3. Why is the value of T set as 1 minute?

4. In Table 1, the authors should mention the unit of RSSI.

5. There are typos in the paper. For example, I have pointed one out here, “in this project,” on line 272.

6. How do the authors determine the requirement of the three services, such as packet size, UL rate, and expected performance of each application?

7. The authors need to add a basic idea/working procedure of the resource allocation (RA) scheme utilized in the comparison.

8.  Avoid expressions such as “In this subsection” on line 283.

Reviewer 2 Report

Introduction section is not state of the art, please add some latest references.

Abstract section is not reflecting the work properly.

English language editing is required.

Proposed resource allocation scheme needs more explanation.  

Reviewer 3 Report

Authors have proposed LoRaWAN harmonization to coordinate multi-services networks based on priority levels where services are classified into three main categories, safety, control, and monitoring.

Essential Questions :

1) How you have decieded 8 number of channels in the simulation parameters ?

In results section can we plot a Total Normalized Throughput Vs Number of Available Channels ?

2) Why you have not compared your algorithim with state-of-the-art approaches available in literature.

3) What about energy consumption ? What is energy cost of data delivery ?

4) What is the impact of the area size on the performance ?

Round 2

Reviewer 1 Report

The authors have addressed all my previous comments satisfactorily. As a result, the quality of the paper has been improved, and now it is up to the standard of the Journal.

In conclusion, this paper is well-written, well-organized, and provides a significant contribution to the field of LoRaWAN communication systems. Therefore, I would recommend this paper for publication.

Reviewer 2 Report

can be accepted.

Reviewer 3 Report

The resubmitted manuscript " Priority-based Resource Allocation Optimization for Multi-Services LoRaWAN Harmonization in Compliance with IEEE 2668" by Yang Weiet al. has been sensibly improved regarding the writing style, the cleanliness of the various sections and answered most of my questions. So the manuscript can be accepted.